# Rainfall-Induced Landslide Prediction Using Machine Learning Models: The Case of Ngororero District, Rwanda

**DOI:** 10.3390/ijerph17114147

**Published:** 2020-06-10

**Authors:** Martin Kuradusenge, Santhi Kumaran, Marco Zennaro

**Affiliations:** 1African Centre of Excellence in Internet of Things, University of Rwanda, Kigali 3900, Rwanda; 2School of ICT, The Copperbelt University, Kitwe 21692, Zambia; santhi.kr@cbu.ac.zm; 3International Centre of Theoretical Physics, Strada Costiera, 11, I-34151 Trieste, Italy; mzennaro@ictp.it

**Keywords:** prediction, rainfall, antecedent rainfall, landslide, random forest, logistic regression

## Abstract

Landslides fall under natural, unpredictable and most distractive disasters. Hence, early warning systems of such disasters can alert people and save lives. Some of the recent early warning models make use of Internet of Things to monitor the environmental parameters to predict the disasters. Some other models use machine learning techniques (MLT) to analyse rainfall data along with some internal parameters to predict these hazards. The prediction capability of the existing models and systems are limited in terms of their accuracy. In this research paper, two prediction modelling approaches, namely random forest (RF) and logistic regression (LR), are proposed. These approaches use rainfall datasets as well as various other internal and external parameters for landslide prediction and hence improve the accuracy. Moreover, the prediction performance of these approaches is further improved using antecedent cumulative rainfall data. These models are evaluated using the receiver operating characteristics, area under the curve (ROC-AUC) and false negative rate (FNR) to measure the landslide cases that were not reported. When antecedent rainfall data is included in the prediction, both models (RF and LR) performed better with an AUC of 0.995 and 0.997, respectively. The results proved that there is a good correlation between antecedent precipitation and landslide occurrence rather than between one-day rainfall and landslide occurrence. In terms of incorrect predictions, RF and LR improved FNR to 10.58% and 5.77% respectively. It is also noted that among the various internal factors used for prediction, slope angle has the highest impact than other factors. Comparing both the models, LR model’s performance is better in terms of FNR and it could be preferred for landslide prediction and early warning. LR model’s incorrect prediction rate FNR = 9.61% without including antecedent precipitation data and 3.84% including antecedent precipitation data.

## 1. Introduction

Landslides and floods are the common natural disasters that strike the northwestern provinces of Rwanda due to its topographical, geological features and climatic profile [1,2]. According to *The National Risk Atlas of Rwanda* report published by the Ministry in Charge of Emergency Management (MINEMA), 42% of areas are classified as moderate to very high susceptible areas to landslides [3]. Statistics indicate that many people lose their lives in different disaster incidences, and the damages of their properties/infrastructure are worth millions of dollars, and much more money is spent on disaster recovery too [4]. Every year, during the rainfall period, landslides affect many people in mountainous regions. These disasters led to the loss of lives and left many homeless and without a livelihood. Since the establishment of an institution in charge of disaster management (MINEMA) in 2010, systematic records indicate that there were 227 deaths and 160 injuries from 2011–2018 (June). Also many houses collapsed and many hectares of crops have been washed away by landslides and floods [5]. Table 1 and Figure 1 summarize the statistics of deaths caused by landslides during the period of 2011–2018 in Rwanda.

In most cases, landslides and floods in Rwanda occur in a cascading manner as debris is dumped into rivers which in turn causes riverine floods [6]. One example is the case of floods and landslides hazards that occurred in 2016 and 2018 where many people died after heavy rainfalls that caused widespread flooding and landslides across parts of Rwanda. At least 222 died in different landslide events since 2011–2018, where victims drowned in floodwater or others died after houses collapsed due to debris movement or landslides caused by heavy rainfall. The type of floods most threatening Rwanda are riverine floods due to its dense river network and large wetlands [7].

The triggering factor of a landslide is rainfall infiltration into the soil, making the groundwater level increase, resulting in a reduction of the shear strength which is also closely related to antecedent rainfall, cumulative rainfall, and rainfall duration [8]. The slope instability increases with high intensity or long duration rainfall but does not relate to rainfall alone. It is affected by other factors such as lithological material, type of soil and depth, the surrounding vegetation, slope inclination or aspect, curvature, altitude, land use patterns, and drainage networks [5,9,10].

Although the rainfall intensity may be the same in different regions, landslides may or may not occur according to the geological and topographical characteristics. Therefore, rainfall intensity cannot be the only cause of landslides [10]. According to the study carried out by MINEMA, the landslide contributing factors have been assigned weights based on the past landslides characteristics in Rwanda: rainfall (20%), slope (20%), altitude (15%), soil type (14%), lithology (10%), land cover (9%), soil depth (7%), distance to the main roads (5%) [7]. Various machine learning models and assessment studies have been carried out on both landslide susceptibility assessments and flood predictions [6,10,11]. The models are built based on the assumption that landslides are more likely to occur in situations similar to those of past landslides [4].

Although there have been numerous researches undertaken on disaster susceptibility assessment and prediction models, the literature about landslides data for Rwanda is scarce. Rwanda lacks efficient early warnings about disasters and hence, the risks of vulnerability by incidences are high [12]. Therefore, there is a need for continuous research on prediction of disasters due to landslides and floods to reduce the risks. Recently, a few studies were conducted on landslide susceptibility in Rwanda, [5] but none of them were about disaster prediction for early warning and risk reduction.

Elsewhere, numerous machine learning techniques have been used to predict landslide occurrence but most of them have been focusing on disaster susceptibility mapping using internal (geological, topographical, environmental) factors without taking into consideration the triggering internal factors such as rainfall [12,13].Those that include rainfall in dataset do not consider antecedent rainfall or vice versa [5,14,15], others use both daily and cumulated previous precipitations without internal conditioning parameters [16]. False negative rate was not considered to measure the performance of the models, yet it is a crucial evaluation metric in landslide predictions.

The main purpose of this research is to improve the performance of the prediction models by including the antecedent rainfall data among other parameters used in the previous studies such as daily rainfall, hill slope angle, soil type, soil depth, and land cover. Another objective is to minimize the incorrect predictions (FNR) as this is a very important metric to be considered since this one counts landslide cases that have not been seen by the prediction model and has a significance in early warning systems. The preferred MLTs are random forest (RF) and logistic regression (LR). These two MLTs were selected among others because of being among the most broadly used in landslide prediction [4] and due to their ability to deal with discrete and continuous data for classification problems. Besides, LR calculates regression coefficients while RF can show up the importance of different parameters, its training speed is high and the computational cost is low [4]. This study is aimed at: (1) to analyze the correlation between rainfall historical data and other topographical and geological factors impacting landslide occurrence in Rwanda and (2) propose a machine learning model for the prediction of this disaster which can be used for early warning. This study is geographically limited to the Ngororero district in Rwanda and the period of study was from 2011–2018.

## 2. Materials and Methods

### 2.1. Area of Study

The district of Ngororero is one of the seven districts of the western province. The district is situated in the northwestern region of Rwanda. The district has a surface area of around 679 km^2^, and is composed of 13 administrative sectors, 73 administrative cells, and 419 villages. The district shares borders with five other districts. The district has a relief characterized by high mountains with very steep slopes that flow into valleys. The altitude varies between 1460 m and 2883 m above sea level, the highest point being on Bweru Mountain situated in Muhanda sector with 2883.4 m of altitude. The average annual temperature is 18 °C which varies with the altitude. The average altitude is 1500 m.

The climate of the region is more of the tropical type with four distinct seasons (short rainy season of October–December corresponding to the agriculture season A; short dry season of January–February; long rainy season of March–June corresponding to the agriculture season B and long dry season of July–September corresponding to the swamp agriculture season C. Rainfall is regular, with a rainfall of 1527.7 mm per year, although irregularities are recorded sometimes with shortages or excess rainfall. Due to much rainfall and topographic structure which is characterized by a steep slope, the district is often affected by landslides (Figure 2) and floods almost every year during the period of heavy rainfall. In the past, some of the victims drowned in floodwater, others died after houses collapsed under the heavy rainfall or landslide [17].

### 2.2. Data Acquisition and Landslide Inventory

#### 2.2.1. Data Collection

Site visits have been conducted in the study area for primary data collection where different landslide-prone areas were visited during the period of 9 months (October 2018–June 2019) to collect GPS coordinates of landslide locations and investigate contributory factors. In addition to the primary data collection, historical data about landslide incidence records have been collected from the ministry in charge of emergency management (formerly called the ministry of disasters management and refugees: MIDIMAR). Since this ministry was established in 2010, landslide data before 2011 could not be found. Therefore, the chosen period for the study is 2011–2018. Daily rainfall data were collected from Rwanda office of Meteorology. Soil type, land cover and slope inclination were collected from different government institutions (Rwanda Land Management and Use Authority, Rwanda Agriculture Board, and Centre for Geographic Information Systems and Remote Sensing (CGIS) at the University of Rwanda. It has been realized that landslides occur under the same or similar conditions as those that caused them in the past [9]. The common internal triggering factor of landslides in Rwanda is the precipitation, but depending on the selected disaster-prone area, the conditioning factors may vary.

#### 2.2.2. Dataset

The internal factors most contributing to the landslide occurrence in Rwanda are slope angle, soil type/texture, soil depth, and land cover while rainfall is the external triggering factor [7]. The daily rainfall data and other parameters collected from stage 1 have been arranged in one datasheet. Geo-topographical data were extracted from Geographic Information Systems (GIS) data. Features of the dataset are described as follows:

#### 2.2.3. Rainfall

Daily historical rainfall data (2011–2018) for the study area were collected from the Rwanda Office of Meteorology. The dataset was made up of daily precipitation from Sovu and Muramba weather stations located in the Ngororero district. The sizes of rainfall data records from each station are 1893 and 2077 entries respectively. Sovu station had 978 rainfall instances while Muramba had 1305 (Figure 3). Because no landslides occurred during the dry seasons in the study area, some non-rainfall days were not included in this dataset. Therefore 3970 instances (rainfall and non-rainfall) have been used.

The two rainfall gauge stations were not sufficient for rainfall data at every location in the study area. To estimate the rainfall in the areas distant from the rainfall gauge stations, the data from neighboring weather stations (Figure 4) were collected and data interpolated using the inverse distance weighting (IDW) method.

The IDW was chosen because it is a commonly used technique for the estimation of missing data in hydrology and geographical sciences [18]. It is based on the functions of the inverse distances in which the weights are defined by the opposite of the distance and normalized so that their sum equals one [19]:(1)$$Z\left(S_{0}\right)=\overset{N}{\underset{i=1}{\sum }}\lambda_{i}Z\left(S_{i}\right)$$
where $Z\left(S_{0}\right)$ represents the interpolated value at point *S*_0_, *Z*(*Si*) represents the observed value at point *Si*, n is the number of observations, and *λi* is the weight. The weights decrease as the distance increases. The weights *λi* can be calculated as follows:(2)$$\lambda_{i}=\frac{d_{i0}^{-p}}{\sum_{j=1}^{N}d_{i0}^{-p}\;}\;\Rightarrow \overset{N}{\underset{i=1}{\sum }}\lambda_{i}=1$$
where *p* is a power and *d*_*i*0_ is the distance between a target and observations. Rainfall data from the two weather stations in the study area and those in three neighboring districts (Figure 4) were used as input into the IDW model and GIS software tool was used for implementation.

#### 2.2.4. Antecedent Rainfall

Analysis indicates that landslides can be triggered by one-day prolonged rainfall or by many consecutive days. From a physical viewpoint, the antecedent rainfall determines the initial water content and matric suction of the soil, but effects of the antecedent rainfall lasts for a certain period [10]. From the rainfall dataset, 5-days antecedent rainfall data have been computed and used as an additional parameter. This parameter has the same size (number of records) as that of daily rainfall.

#### 2.2.5. Slope

One of the most important landslides’ causal factors is slope [20]. Increase in the slope decreases its stability if the soil depth is sufficient [21]. According to the “National risk atlas of Rwanda” by the ministry in charge of emergency management, the slope is classified into ranges from 0 to 10 according to the steepness angle [7].

#### 2.2.6. Soil Type

Based on the grain-size distribution analysis and prevailing range of particle sizes, soils in the study area are classified into three categories: sand, silt, and clay [7].

#### 2.2.7. Soil Depth

This was considered as one of the landslide causal factors. Three classes of soil depth were used in the dataset according to MINEMA soil depth classification [7].

#### 2.2.8. Land Cover

Various past studies have pointed out that permanently covered lands are more protected against landslides than non-covered ones. According to their potential influences, there are six main types of vegetation in Rwanda [7], but three are dominant in the area of study as shown in Table 2.

All the data in Table 2 have been collected as secondary data. Slope and soil depth are continuous while on the other hand, the soil type and land cover are categorical. Both data were obtained from the source (government institutions) with the assigned scores (claimed to be FAO scores) for the purpose of getting qualitative data as defined in some official documents [7]. The standard scores were used as categorical data during model training process. Figure 5 shows the geographical characteristics (as defined in the area of study).

The complete dataset was made of 3970 daily rainfall × 5 slope categories × 3 soil type categories × 3 soil depth categories × 3 land cover categories = 535,950 dataset records.

#### 2.2.9. Landslide Incidences

Data about landslide events with their respective date of occurrences were collected from the ministry in charge of emergency management (MINEMA) and were recorded as either zero (0) for “no landside event” or as one (1) meaning “landslide event”. The full dataset used in this study is available online [22]. The datasheet was converted to the comma-separated value (CSV) format for the reason of being used in the machine learning algorithm. At this stage, the missing values were handled using the fill in missing values with the median technique and labels removed from features.

#### 2.2.10. Splitting Dataset into a Training and a Test Dataset

The dataset was divided into training set and test set at different proportions.

#### 2.2.11. Training and Testing the Models

The training dataset was used to train two machine learning models and predictions were done using the test dataset. Test labels were used for the accuracy calculation.

#### 2.2.12. Results Analysis

Based on the performances, the best model will be recommended to be implemented for early warning system.

### 2.3. Machine Learning Models

This study aims to apply two machine learning techniques and choose the best performing one for predicting landslide incidences in Rwanda, which can be used for early warning to reduce risks. Two classification models that have been selected to accomplish this research area are random forest (RF) and logistic regression (LR). The simplicity or complexity of each MLT differs from one to another. Hyper-parameter tuning has been done to optimize the performance of each. These classification models were chosen because this research is dealing with the binary classification problem. The complete research methodology is indicated in Figure 6.

#### 2.3.1. Random Forest

Random forest (RF) was first introduced by Breiman [23] as an ensemble learning algorithm. It is a commonly used machine learning technique for data prediction for both classification and regression problems [23,24,25]. It is called random forest because of randomness in the selection of features on each decision tree and also training samples from the dataset [26]. It operates by building up several decision trees from random subsets of dataset during the training time to make up a relationship between features and labels [27]. Decision or classification tree (Figure 7) is a tree-shaped diagram which is a basic structure block of RF used to decide a route of action. RF consists of the construction of classification rules (tree) from bootstrap samples of the dataset. The decision of the final classification is computed from the majority vote of multiple classification trees.

The bigger number of trees in the model plays a role in improving the accuracy and predictive capability [29] and requires defining the number of trees (T) and the number of predictive variables (m). RF can work with both categorical and numerical data [20]. RF was chosen because of its power to deal with mixed variables (categorical and numerical), over-fitting risks reduced, efficiently working on large databases with high accuracy, robustness with less training time, estimates missing data, and feature importance indication [30].

#### 2.3.2. Logistic Regression

Logistic regression (LR) is a machine learning algorithm that was initially used in biological sciences (during the early 20th century) and later in other areas. It is used in solving classification problems by measuring the relationship between independent variables and categorical dependent variables. The technique has become one of the most used in landslide susceptibility assessment [20]. It is in three categories: Binary (dichotomous or has two possible classes coded as 0/1: absence/present), Multinomial (more than two categories without ordering), and ordinal LR (more than 2 categories with ordering). In its simplest form LR model can be expressed using a sigmoid function to makeover regression value in a range of −∞ to +∞ producing a probability between 0 and 1 for non-landslide and landslide respectively with 0.5 being threshold [31]:
(3)$$P\left(Y=1\right)=\frac{1}{1+e^{-y}}$$ where *P* is the probability of event occurrence and *y* is the dependent variable (landslide or no-landslide) and is defined as:
(4)y=β0+β1X1+βX2+⋯βnXn where *y* is dependent variable and *X*1, *X*2, … and *X*n are explanatory variables. Then:
(5)$$P\left(Y=1\right)=\frac{1}{1+e^{-\left(\beta 0+\beta 1X1+\beta 2X2+\ldots \beta nXn\right)}}=\frac{e^{\beta 0+\beta 1X1+\beta 2X2+\ldots \beta nXn}}{1+e^{\beta 0+\beta 1X1+\beta 2X2+\ldots \beta nXn}}$$ where β0 is an intercept of the model and *β*1, *β*2,….*β**n* are regression coefficients that measure the contribution of *x*1,*x*2, …*x**n* to the prediction model. With LR mathematical equation can be generated from regression coefficients and intercept, hence the probability of landslide incidence can be computed.

### 2.4. Re-Sampling

One of the challenges of various machine learning models is that their performance is affected by imbalanced data in the training dataset. To deal with this problem, three re-sampling techniques have been used: oversampling, under-sampling and synthetic minority oversampling technique (SMOTE) and the later found to be the best. SMOTE was initially proposed in 2001 by Nitesh [31]. It uses the nearest neighbor’s algorithm to create new instances of minority categories by forming a convex combination of neighboring instances.

### 2.5. Preliminary Analysis Using Exploratory Data Analysis (EDA)

Exploratory data analysis (EDA) was used to get patterns and the relationship between landslide incidences and various parameters [31]. The scatter plot matrix tool was used as methods of EDA and helped to identify the relationship among variables.

### 2.6. Models Evaluation

Proper evaluation of the performance of the model is a crucial aspect of predictive modelling. Wide ranges of performance metrics are available for classification and regression models. A common metric for binary classification is the area under the receiver operating curve (ROC). The most widely used evaluation metric is confusion metric, which distinguishes measures between errors and measures overall accuracy or percent correct classification. In this study, the statistical index-based method and ROC have been used to evaluate performance of the models. In this regard, the table of summary of classification errors (confusion matrix) was used to calculate the following evaluation indexes:(6)$$Accuracy=\frac{TP+TN}{TP+FN+TN+FP}$$
(7)$$FPR=\frac{FP}{FP+TN}$$
(8)$$Recall=\frac{TP}{TP+FN}$$
where Accuracy is the overall performance of the classifier, false positive rate (FPR) is the rate of incorrect predictions. *TP* or Recall denotes the true positive (correctly predicted incidences), *TN* represents true negative (correctly predicted as negative) [32]. *FP* is the false positive (predicted occurrences but did not occur), *FN* is the false negative (predicted negative but in reality it was positive). These performance metrics were selected based on their high impact on predicted landslide. Moreover, the metrics are most relevant for the proposed approach as the model will be combined with wireless sensor network (WSN) that will be used as early warning system to alert people about landslide occurrence and reduce risks.

The receiver operating curve is one of the most widely used models’ performance tools for classification problems. It summarizes all the confusion matrices that each threshold produced by plotting FPR (1 - sensitivity) on X-axis vs. Sensitivity or Recall (TPR) on Y-axis [33]. ROC curve makes it easy to identify the best threshold to make a decision. The area under the curve (AUC) is also used to compare two or more ROC curves. AUC is used to decide which classification method is better. The following equation is used to calculate AUC [34]:(9)$$AUC=\frac{\left(\sum^{\;}TP+\sum^{\;}TN\right)}{P+N}$$
where *P* is the total number of positives (landslides), *N* is the total number of negatives (No landslides).

## 3. Results

### 3.1. Preliminary Analysis: Correlation Among Features Used in the Models

As shown in Figure 8, there is no specific threshold for 1-day rainfall inducing landslides. Rainfall from 0 mm could cause landslide occurrence depending on how much it rained in the previous days. As indicated by graphs, very few landslide hazards occurred for antecedent rainfall less than 50 mm. In most cases, landslides occurred when previous cumulative precipitation was close to 100 mm.

Also, as indicated in Figure 9, many landslides occurred in hilly areas where the slope inclination is 25–45° (category 6 and 10) and very few landslides occurred in the area where the slope is less than 15° (category 1 and 4). The land covered by the forest resists landslide hazards but, also the low number of landslide incidences in that area can be justified by the low dominance of that category. Agricultural land is the most affected by slope failure while the forest is less affected.

The soil under the category of silt was much affected by landslides while areas of sand have few incidences just because of physical properties of this soil particles but also because not much of this type of soil is found in the district. Figure 10 points out that few landslides incidences occurred in the areas where the soil depth is thin (less than 50 cm) while more occurred in the regions with thick soil (depth bigger than 50).

### 3.2. Models Results

The performance evaluation of the classifiers was done by using five different metrics including Recall (or percent correlation classification), accuracy, FPR, FNR, and the ROC-AUC. Four internal parameters and one external (triggering factor) have been used by the models. Initially, the models were trained by using 1-day rainfall followed by training models with the inclusion of the cumulated 5-days antecedent precipitation as a new parameter into the dataset.

Initially, the models have been trained using 5-fold cross validation and later tested using different train/test ratios (0.8 and 0.2, 0.75 and 0.25, 0.7 and 0.3, 0.65 and 0.35, 0.6 and 0.4, 0.55 and 0.45) for determining which provide better results. By applying 5-fold cross validation, the variance is very low, as indicated by the insignificant standard deviation for both models which is 0.0003 and 0.0004 for RF and LR, respectively (Table 3). The overall prediction accuracy was 95.30% (RF) and 94.63% (LR) by using cross validation while 0.6 and 0.4 ratio provided better results with an accuracy of 95.35% for RF and 0.55 and 0.45 was the best ratio for LR. On the other hand, if the antecedent rainfall is included as the new feature in the models, the overall accuracy was 98.74 and 98.79% (RF, LR respectively), the best ratio for RF cross validation was 0.65 and 0.35 with an accuracy of 97.67%, while 0.60 and 0.40 ratio looks to be better in LR model with 98.40% accuracy.

It has been observed that there is no significant difference in applying different train/test ratios as indicated by the standard deviation (0.0001). However, the ratio of 65% of the dataset was adopted as training, while 35% was used as a test sample. This ratio was preferred for the reason that LR 0.65 and 0.35 is the best ratio for incorrect predictions (FNR) and for maintaining the same ratios to both models. The overall accuracy could not be considered as only one metric to judge the performance of the models because of the imbalance of dependent values in the dataset. Therefore, to assess the performance of the models, other metric indexes have been used. FN and FP were used to evaluate the models in terms of incorrect predictions such as landslide cases which have been considered as no landslides, yet landslides occurred. In addition to the prediction results presented in Table 4 RF generated features of importance indicating how much each conditioning factor contributes to the landslide incidence. As the triggering factor, rainfall has the highest contribution rate on landslide occurrence. If 1-day rainfall data are used (without antecedent precipitation), rainfall contributes 40.49%, slope 32.74%, soil type 16.48%, land cover (land use) 8.46%, and soil depth 1.80% (Figure 11a).

Upon including antecedent cumulated rainfall, the most contributing factor was an antecedent rainfall at the rate of 40.64%, followed by the rainfall on the day of landslide occurrence contributed 22.91%, slope with 22.50%, soil type 7.87%, land cover 5.21%, and soil depth 0.84% (Figure 11b).

Also, LR model provided the intercept and coefficients of variables form which the mathematical equation establishing the relationship between labels (dependent variables) and features (independent variables) was derived. Variables’ coefficients reveal which parameter has much or low impact on landslide occurrence as indicated by positive or negative values in Table 5.

By using Equations (3) and (4) and values of intercept and coefficients in Table 5, the probability of landslide occurrence can be expressed by the following equation:(10)$$P(Y=1)=\frac{1}{1+e^{-y}}=\frac{e^{y}}{1+e^{y}}$$
where P is the probability of landslide occurrence and:(11)$$\begin{array}{ll} Y=-7.06+ & 1.32\mathrm{rf}\;+2.47\mathrm{Ar}-9.34{\mathrm{Sl}}_{0-10}-9.29{\mathrm{Sl}}_{11-15}+1.2{\mathrm{Sl}}_{16-20}+3.87{\mathrm{Sl}}_{21-25}\\ & \;+6.49{\mathrm{Sl}}_{26-45}-2.9{\mathrm{St}}_{\mathrm{clay}}-5.39{\mathrm{St}}_{\mathrm{sand}}+1.23{\mathrm{St}}_{\mathrm{silt}}-4.10{\mathrm{Sd}}_{<0.5}\\ & \;-1.85{\mathrm{Sd}}_{0.6-1.0}-1.10{\mathrm{Sd}}_{>1.0\;}-4.37{\mathrm{Lc}}_{\mathrm{forest}}-1.12{\mathrm{Lc}}_{\mathrm{agri}}\\ & \;-1.56{\mathrm{Lc}}_{\mathrm{open}} \end{array}$$
where −7.09 is the intercept, Rf is one day rainfall, Ar is antecedent rainfall, Sl is a slope in degrees (0–10, 11–15, 16–20, 21–25, 26–45) respectively, St is the type of soil (clay, sand and silt), Sd is soil with (>0.5 meter, 0.5–1.0 meter, more than 1.0 meter) depth and Lc is land cover (forest, agriculture and open land).

## 4. Discussion

Prediction of landslide occurrence and early warning systems are the primary keys for the awareness and preparedness for risk reduction from this type of disaster. Different prediction models and warning systems (both local and regional) are available for this purpose [35]. The most popular machine learning models that have recently been used for landslide prediction are namely random forest [20,25,36], artificial neural network [37,38], support vector machine [4,25], logistic regression [36,38,39], etc. The results obtained from these different studies were good depending on the variables used and metrics used to evaluate their performance. In some studies, only internal factors were used to determine landslide susceptibility, some others have used rainfall, while a few of them included antecedent precipitation to determine its impact on the disaster occurrence. In this study, we preferred to use RF and LR for the reason explained earlier and more particularly to be applied in Rwanda as, to the best of our knowledge, there is no such study carried out on this territory. The use of both internal (geological and morphological) factors together with external (triggering) factor (rainfall: antecedent & current) led us to the better performance of the models compared to when either one of these factors is not considered. The results from a comparative analysis of one-day rainfall and 5-days antecedent cumulative rainfall before landslide occurrence revealed that antecedent cumulative rainfall has more impact on landslide occurrence than one-day rainfall (Table 6). This is because the shear strength of the soil is degraded by rainfall and continuous if the interval between two or more rainfall events is short (few hours) but also depending on the hydraulic conductivity of the soil [40].

All performance metrics improve when the antecedent rainfall (Ar) is used for both models. For correct predictions, RF improves 10.58 and 3.73% for recall and specificity respectively while LR improves 5.77 and 4.46% on the same metrics. Incorrect predictions improve in the same manner as the correct predictions as ones are the opposite of the other ones. FN is a very important parameter to consider in the case of landslide prediction as its low value means few incidence cases were not predicted. In this case, LR performs better than RF as Table 6 shows that only 3.84% cases were not predicted.

The receiver operating curve was used to identify the performance of the models at different classification threshold and the AUC was used to compare two models by including or not including antecedent cumulated rainfall. Considering ROC-AUC, RF performs with AUC = 0.973 (Figure 12a) if only daily rainfall is considered as landslide triggering factor and AUC = 0.995 by including antecedent rainfall as additional landslide triggering factor (Figure 12b). This means that antecedent cumulated rainfall implicates better performance of the model to the rate of 2.2%. Likewise, Ar improved the LR model as before the inclusion of previous cumulative precipitations AUC was 0.979 (Figure 12a). However, after including the Ar, AUC was equal to 0.997 (Figure 12b), hence this parameter contributed 1.8% to the model prediction.

Both models indicated that 5-days antecedent precipitation has a high impact on the occurrence of landslides in the study area. Table 5 shows that the features most triggering landslide incidences are rainfall (one-day, 5-days antecedent) as indicated by their coefficients of 2.47 and 1.32, respectively. Among the internal factors, the slope is the most affecting the occurrence of the disaster. The magnitude of the impact increases as slope increases or vice versa. For low slopes (less than 15 degrees coefficients are negative while the probability of landslide incidence is very high on slopes of 25–45 degrees with a coefficient of 6.49. This means that high slope zones are prone to landslides and people should not reside in such areas to reduce risks. The silt soil, the soil of more than 1m deep, and agriculture land are the most vulnerable to the landslide (Table 5).

Table 6 shows that both models performed well in terms of different performance indicators such as accuracy, errors, and AUC. The use of antecedent precipitation made the models predict better. We also compared the performance of these prediction models with other recently established models using the same machine learning algorithms, [26,41,42,43,44] and the ones used different models such as artificial neural networks (ANNs) [26], support vector machine (SVM) [26], event-class predictor [45], and rotation forest with alternating decision tree (RFADT) [41]. The comparison of the models was done using AUC which seems to be widely used by researchers and FNR which has been used by a few. The ANN proposed by Wang et al. [26] performed better compared to the RF and LR models proposed in this study before the use of antecedent rainfall data, but after including the antecedent rainfall data in the dataset both RF and LR models predict better with good AUC results compared to ANN and other models. Thus, adding antecedent rainfall data has made the RF and LR models proposed in the current study achieve good results (Figure 13).

A comparative analysis has also been done with other models on incorrect predictions. Figure 14 indicates that the model proposed by Utomo et al. [36] is the best to minimize false negative errors up to 1.60%, while the LR in our study can minimize FNR up to 3.84%. On the other hand, our proposed LR can minimize FP errors up to 1.61% against 2.01% of Utomo et al. [36].

Comparing the two proposed models RF and LR of the current study among themselves, there is a slight difference in the prediction parameters, like for instance if ROC-AUC is considered both models are good, as indicated in Figure 12, but in terms of prediction errors, LR is considered the best because its FNR is 3.84% against 4.80% of RF. This evaluation parameter (FNR) should be highly considered because it indicates how many landslide cases were predicted as NO landslides, which can be a dangerous outcome. Minimizing the ratio of false negatives is crucial for better performance of the system and disaster risk reduction. In this regard, logistic regression is the best model to be used due to its performance in terms of error reduction (false negatives). Besides error reduction, LR performs faster than RF which is another important aspect that LR to be considered in the early warning system as the delay is reduced. Though 100% perfect prediction is not achievable through these modelling approaches, the priority is to minimize the false negatives (warning that there will be no landslide yet landslides occur) because it has a negative impact on lives rather than false alarms (warning that there will be a landslide while is false).

## 5. Conclusions

In this research two approaches, random forest (RF) and logistic regression (LR), were applied to analyze the rainfall data along with other external and internal factors to develop a prediction model for landslide incidences for an early warning system. Performance parameters such as ROC-AUC, error rate (TP and FN) have been used to evaluate the best prediction models. The results prove that the prediction performance of these two models are better than those established in other research studies.

Results from this study revealed that landslides are triggered due to the too much daily (or low intensity prolonged) rainfall, but in most cases they occurred after a few consecutive (like 2–5) days of precipitation. This was proved by the impact of antecedent rainfall on disaster occurrence as marked by prediction models used in this study. Both models indicated that 5-days antecedent precipitation has a high impact on the occurrence of landslides in the study area. In addition to the rainfall data, other parameters have been utilized to assess their impact on the disaster. The slope of hills is the most internal parameter affecting disaster occurrence after rainfalls (external factor), meaning that the areas of high slope angle are more susceptible to landslides than the regions where the terrain is almost a plateau. Agricultural land or non-protected land is the most susceptible to landslide occurrence while land covered by forest had few incidences. As previously discussed, landslides are triggered by current rainfall but very correlated with consecutive precipitation before the day of incidence. This is because the soil strength reduces with water content depending on the type of soil material and gets strong again during the evapotranspiration process or the sunny season. Based on the results, the logistic regression proves to be the best approach to be used for landslide prediction and early warning. LR model’s incorrect prediction rate FNR is 9.61% without including antecedent precipitation data and is 3.84% after including the antecedent precipitation data. Therefore, LR model can be used for the early warning system.

Instead of antecedent cumulated rainfall, in our future work, we will study the correlation between landslide occurrences, rainfall and soil moisture level on different types of the soil which will be gathered by sensors and through WSN, landslides will be predicted and citizens warned earlier. The prediction capability combined with the Internet of Things (IoT) where rainfall gauges can be used to capture rainfall intensity in real-time and soil moisture sensors used to measure the water content in the soil will be used for rescuing citizens from landslide disasters.

## Figures and Tables

**Figure 1 ijerph-17-04147-f001:**
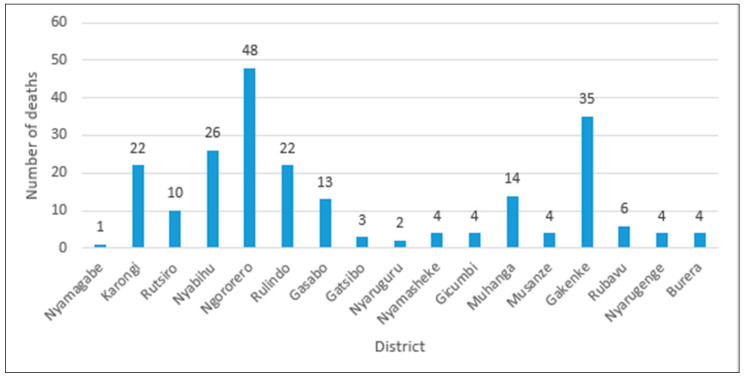
Number of deaths caused by landslides per district.

**Figure 2 ijerph-17-04147-f002:**
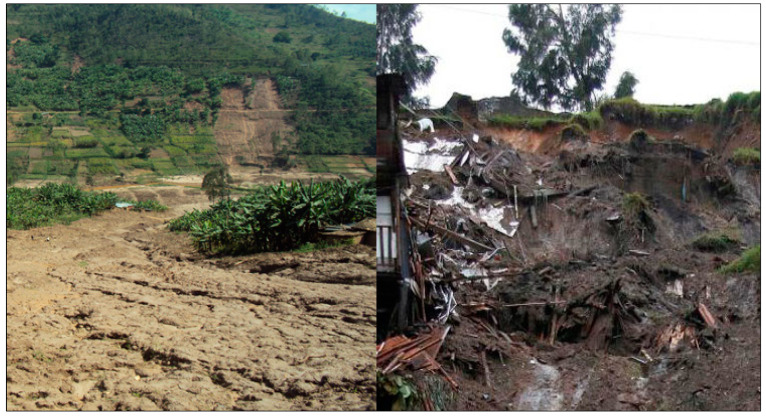
Photographs of some landslides in the study area.

**Figure 3 ijerph-17-04147-f003:**
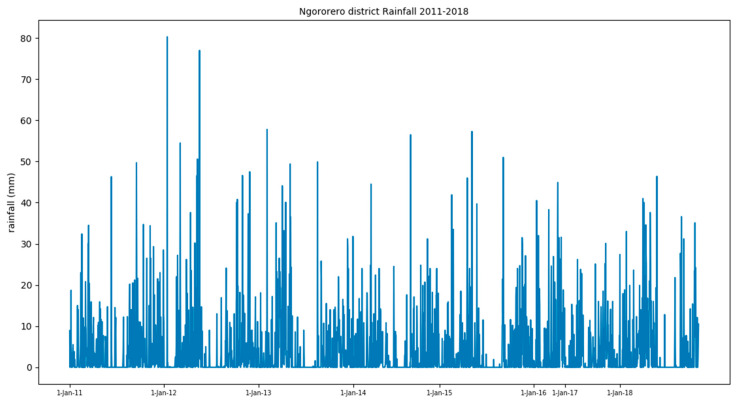
Rainfall in the study area.

**Figure 4 ijerph-17-04147-f004:**
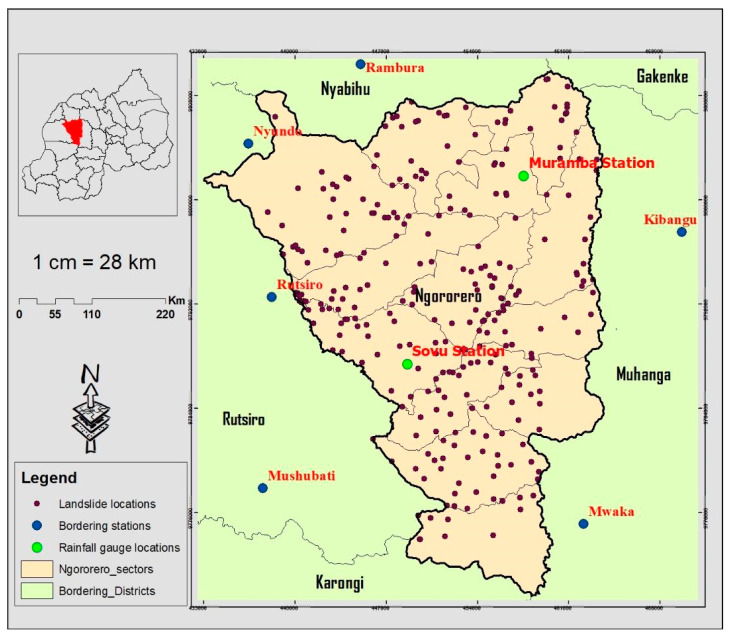
Landslide inventory in the study area and rain gauge stations.

**Figure 5 ijerph-17-04147-f005:**
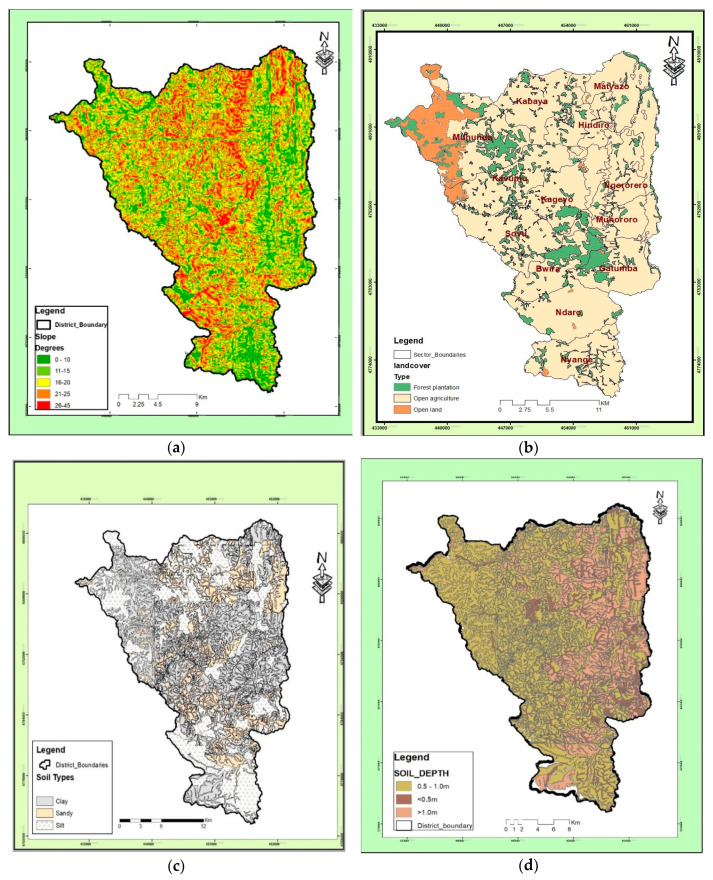
Geographical characteristics in the study area: (**a**) slope, (**b**) land cover, (**c**) soil type, (**d**) soil depth.

**Figure 6 ijerph-17-04147-f006:**
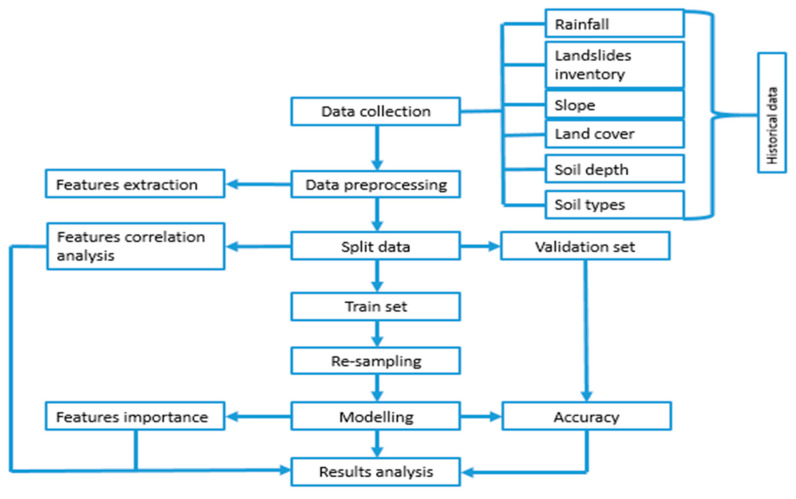
Proposed landslide prediction flow chart.

**Figure 7 ijerph-17-04147-f007:**
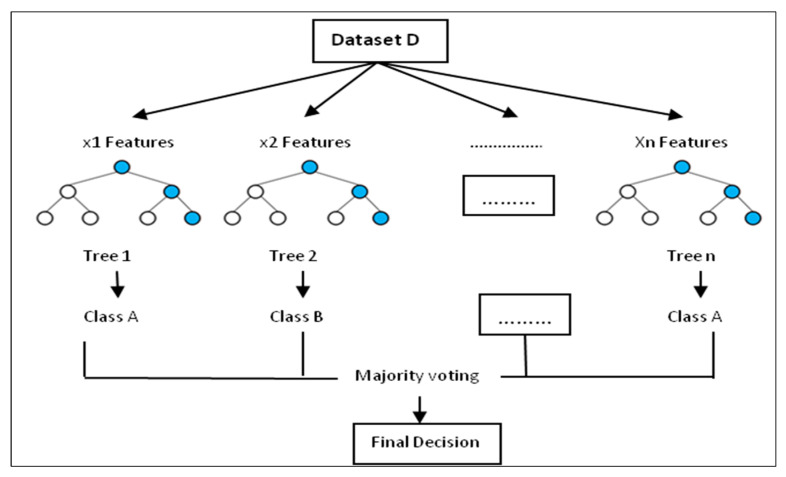
Random Forest decision tree [28].

**Figure 8 ijerph-17-04147-f008:**
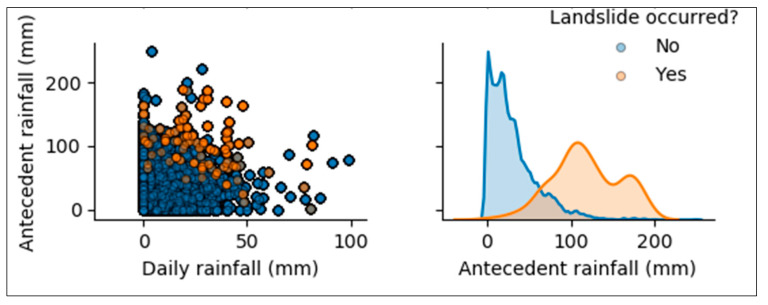
Correlation between landslide and rainfalls.

**Figure 9 ijerph-17-04147-f009:**
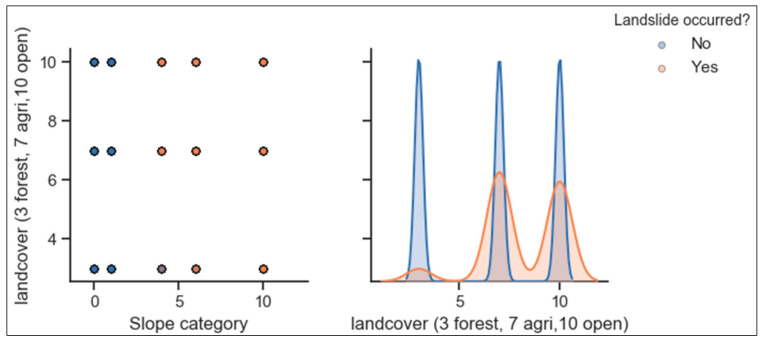
Correlation analysis between landslides, slope and land cover.

**Figure 10 ijerph-17-04147-f010:**
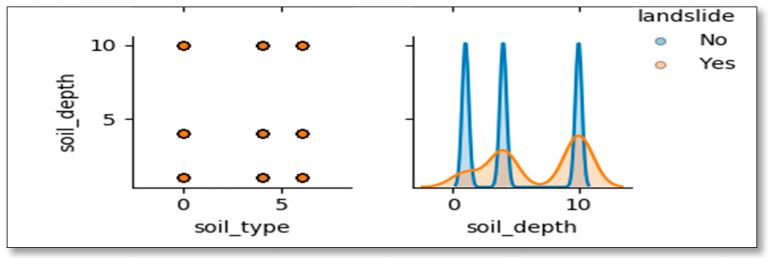
Correlation analysis between landslide and soil.

**Figure 11 ijerph-17-04147-f011:**
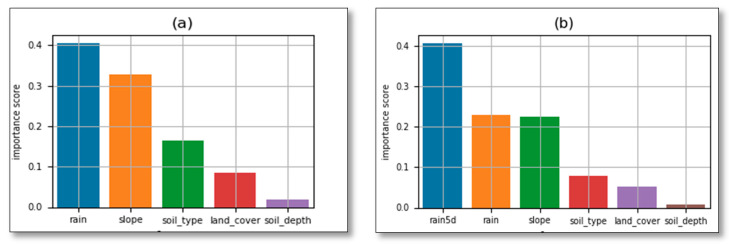
RF features importance (**a**) without antecedent rainfall (**b**) with antecedent rainfall.

**Figure 12 ijerph-17-04147-f012:**
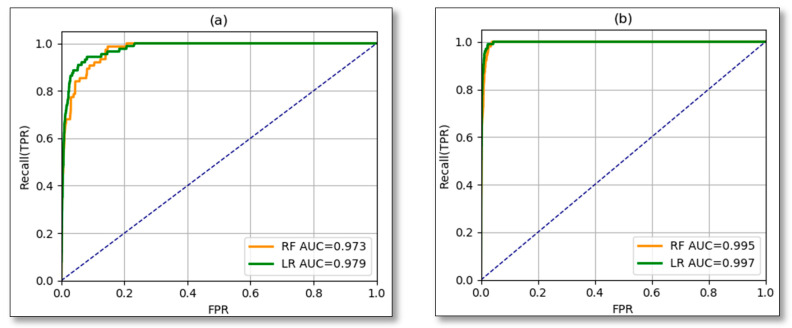
ROC-AUC (**a**) with no antecedent rainfall, (**b**) 5-days antecedent rainfall taken into consideration.

**Figure 13 ijerph-17-04147-f013:**
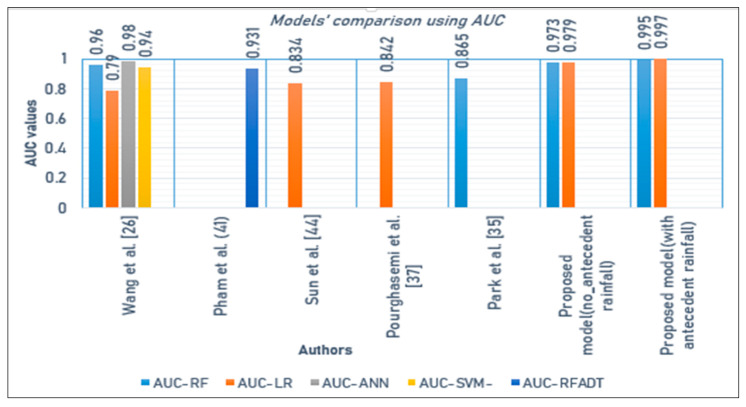
Comparison of the models results (using AUC) vs. other studies.

**Figure 14 ijerph-17-04147-f014:**
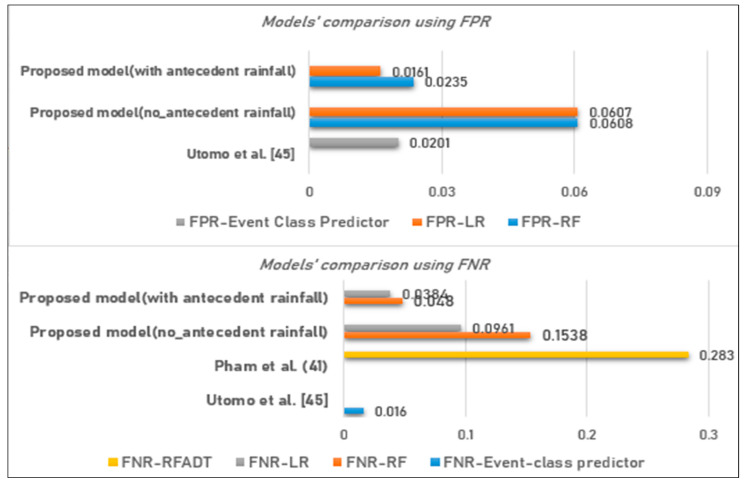
Comparison of the models results (false predictions) vs. other studies.

**Table 1 ijerph-17-04147-t001:** Number of deaths caused by landslides in Rwanda (2011–2018).

Year	2011	2012	2013	2014	2015	2016	2017	2018	Total
Number of death	19	14	31	0	10	64	7	77	222

**Table 2 ijerph-17-04147-t002:** Summary of classification of key parameters in the study area and their standardized scores used in the model.

Slope		Soil				Land	
Slope Angle (Degree)	Score	Soil Type	Score	Soil Depth (cm)	Score	Land Cover	Score
0–10	0	Clay	0	<50	1	Forest plantation	3
>10–15	1	Sand	4	>50–100	4	Agriculture	7
>15–20	4	Silt	6	>100	10	Open land	10
>20–25	6						
>25–45	10						

**Table 3 ijerph-17-04147-t003:** Models’ overall accuracy.

1-Day Rainfall (Without Antecedent Rainfall)						5-Days Antecedent Rainfall Included				
Fold	Accuracy (using cross-validation)		Accuracy (using train/test ratios)			Accuracy (using cross-validation)		Accuracy (using train/test ratios)		
	RF	LR	Ratio	RF	LR	RF	LR	Ratio	RF	LR
1	0.9530	0.9466	0.80 & 0.20	0.9414	0.9322	0.9875	0.9877	0.80 & 0.20	0.9742	0.9836
2	0.9536	0.9466	0.75 & 0.25	0.9398	0.9330	0.9871	0.9882	0.75 & 0.25	0.9759	0.9835
3	0.9531	0.9459	0.70 & 0.30	0.9399	0.9352	0.9876	0.9878	0.70 & 0.30	0.9744	0.9838
4	0.952	0.9457	0.65 & 0.35	0.9394	0.9392	0.9870	0.9879	0.65 & 0.35	0.9767	0.9838
5	0.9530	0.9467	0.60 & 0.40	0.9469	0.9413	0.9876	0.9879	0.60 & 0.40	0.9740	0.9840
Av.	0.9530	0.9463	0.55 & 0.45	0.9409	0.9415	0.9874	0.9879	0.55 & 0.45	0.9744	0.9837
Std	0.0003	0.0004		0.0028	0.0041	0.0002	0.0001		0.001	0.0001

**Table 4 ijerph-17-04147-t004:** Performance results.

Performance Metric	1-Day Rainfall, Antecedent Rainfall Excluded (%)		5-Days Antecedent Rainfall Included (%)	
	RF	LR	RF	LR
Recall (TPR)	84.61	90.38	95.19	96.15
Specificity (TNR)	93.91	93.92	97.64	98.38
False Positive Rate (FPR)	6.08	6.07	2.35	1.61
False Negative Rate (FNR)	15.38	9.61	4.80	3.84

**Table 5 ijerph-17-04147-t005:** LR intercept and coefficients of parameters.

Intercept	Daily Rainfall	Antecedent Rainfall (5-Days)	Slope	Soil Type	Soil Depth	Land Cover
−7.06	1.32	2.47	−9.34	−2.90	−4.10	−4.37
			−9.29	−5.39	−1.85	−1.12
			1.20	1.23	−1.10	−1.56
			3.87			
			6.49			

**Table 6 ijerph-17-04147-t006:** Models’ performance summary (various metrics).

	Random Forest			Logistic Regression		
	**Correct Predictions (%)**					
**Performance Metric**	**Without Antecedent Rainfall**	**With Antecedent Rainfall**	**Improvement (%)**	**Without Antecedent Rainfall**	**With Antecedent Rainfall**	**Improvement (%)**
Recall (TPR)	84.61	95.19	10.58	90.38	96.15	5.77
Specificity (TNR)	93.91	97.64	3.73	93.92	98.38	4.46
	**Incorrect Predictions (%)**					
False Positives	6.08	2.35	3.73	6.07	1.61	4.46
False Negatives	15.38	4.80	10.58	9.61	3.84	5.77

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
