# Peer review of "Rainfall-Induced Landslide Prediction Using Machine Learning Models: The Case of Ngororero District, Rwanda"

_ijerph, 2020, doi:10.3390/ijerph17114147_

Round 1

Reviewer 1 Report

Dear authors. Your article has potential but it cannot be published in its current form. Please consider the comments below which are aimed at helping improve the quality of your work.

Presentation issues: your article needs a substantial language review. There are also issues related to punctuation marks and broken cross references.

Scientific issues: there is a considerable number of important references that belong to the subject of interest of your research but were not cited in your work. Also, two things are not clear in the text: (1) the novelty of your work and (2) the limitations of your work.

General comments:

  • What's the novelty of your work? Please make it clear in the text.
  • Regarding the database you used in this research, have you made it available for the wide public? If so, please include a link to the database and include it in your references. If not, please upload them to an online platform and then cite it in your text. It gives your work more credibility since the experiments can be repeated by anyone who might be interested.
  • Please explain the values of the 'standardized scores' used in the model. How were the values obtained? If they were arbitrarily assigned to obtain a quantitative parameter from a qualitative one, then write it on the text.
  • Where are the rainfall gauges located (those from Sovu and Muramba)? Maybe a map would clarify that. And how did you obtain the rainfall data to other locations? Interpolation? Please clarify that and explain it in detail in the text.
  • The authors should compare their results to other studies performed in the same area (using other methods) and to other studies performed with the same methods (in other areas) to provide a comprehensive to the reader.

Author Response

Dear Reviewer, 

Please receive our attached feedback to the comments provided to us while reviewing our manuscript

Regards

Reviewer 2 Report

I would like to congratulate the authors on an interesting paper and would recommend publication with only a few minor changes. First, some parts of the paper require a more thorough revisit regarding the English language, for example in lines 22, 33, 34, 39, 45, 51, 62, 64, 73, 99, 130, and many other. Please carefully double check the whole document.

Other minor issue are:

53-55: I would suggest you stick to the sme dates that are ised in the figures above

60/61: Lithology and lithological material is the same thing

133-137: Repetion of factors, e.g. soil depth etc.

157: Can you explain in one sentence why those intervals for slope angle are chosen.

159: Can you be more specific in your soil types, by which you seem to mean soil texture. It's not all sand or silt or clay, but some combination.

190: Yes, please include all references

Figure 8: Is there a source for this figure. If not, you may want to modify it such that the trees do not all look exactly the same

250: What kind of business?

255-259: Please rewrite for clarity (mostly an English language issue)

260-272: see previous comment

283: That should be fig. 9

Figures 9-11: They are a little unclear to me. Please redo them with better indication of units on the axes. They would also bebefit from better labels and a more comprehensive caption.

Author Response

Dear Reviewer,

We would like to thank you for reviewing our manuscript and constructive comments on the work. We are pleased to submit to you our revised version for your more support. See attachement

Regards
